# Control of Gene Expression by Proteins That Bind Many Alternative Nucleic Acid Structures Through the Same Domain

**DOI:** 10.3390/ijms27010272

**Published:** 2025-12-26

**Authors:** Alan Herbert

**Affiliations:** Discovery, InsideOutBio, Charlestown, MA 02129, USA; alan.herbert@insideoutbio.com

**Keywords:** flipons, Yamanaka factors, anti-oxidant responses, transcription factors, alternative DNA structures, Z-DNA, G-quadruplex, bZIP, bHTH, oncogene

## Abstract

The role of alternative nucleic acid structures (ANS) in biology is an area of increasing interest. These non-canonical structures include the Z-DNA and Z-RNA duplexes (ZNA), the three-stranded triplex, the four-stranded G-quadruplex (GQ), and i-motifs. Previously, the biological relevance of ANS was dismissed. Their formation in vitro often required non-physiological conditions, and there was no genetic evidence for their function. Further, structural studies confirmed that sequence-specific transcription factors (TFs) bound B-DNA. In contrast, ANS are formed dynamically by a subset of repeat sequences, called flipons. The flip requires energy, but not strand cleavage. Flipons are enriched in promoters where they modulate transcription. Here, computational modeling based on AlphaFold V3 (AF3), under optimized conditions, reveals that known B-DNA-binding TFs also dock to ANS, such as ZNA and GQ. The binding of HLH and bZIP homodimers to Z-DNA is promoted by methylarginine modifications. Heterodimers only bind preformed Z-DNA. The interactions of TFs with ANS likely enhance genome scanning to identify cognate B-DNA-binding sites in active genes. Docking of TF homodimers to Z-DNA potentially facilitates the assembly of heterodimers that dissociate and are stabilized by binding to a cognate B-DNA motif. The process enables rapid discovery of the optimal heterodimer combinations required to regulate a nearby promoter.

## 1. Introduction

The flip from B-DNA to a noncanonical structure has a physicochemical correlate. In many cases, it is characterized by the rotation of a nucleobase from the *anti*- to the *syn* conformation, where the base is directly above the sugar rather than pointing away, as occurs in the anti-conformation [1,2,3]. The energy cost of flipping to a syn orientation is lower for purine bases than for pyrimidines, as steric clashes can occur between pyrimidine and sugar adducts, and is lowest for guanosine. Non-canonical DNA structures exhibit a characteristic combination of syn and anti bases. These motifs are most often found in repeat sequences. Preferred sequences for forming Z-DNA and Z-RNA (collectively called ZNA) are alternating purine-pyrimidine repeats [4], while the tetrads that form G-quadruplexes (GQ) are based on G-repeats [5], triplexes (TPX) on polypyrimidine [6,7,8], with i-motifs formed by polycytosine tracts [9].

The evidence supporting a biological role for flipons is most complete for ZNAs [10]. Z-flipons are an integral component of the innate immune system. ADAR and ZBP1 modulate pathways through the structure-specific Zα domain, which docks with nanomolar affinity to the left-handed conformation of both RNA and DNA [11,12,13]. Loss-of-function (LOF) human variants of the ADAR Zα domain are causal for Aicardi-Goutières Syndrome Type 6. These phenotypes are supported by mouse studies that show the variants impair the negative regulation of type I interferon responses [14,15,16].

In contrast, the binding of ZBP1 to ZNA activates immune responses during viral infection, leading to inflammatory cell death (referred to as necroptosis). In other contexts, ZBP1-dependent caspase activation leads to either the inflammatory outcome called pyroptosis, or the immunologically silent form of cell death called apoptosis. ZBP1 is rapidly induced by interferon. Besides regulating interferon responses, ADAR1 squelches the activation of cell death by ZBP1 by competing for ZNAs [17,18,19,20,21,22,23,24,25,26,27,28,29].

Roles for Z-DNA in transcription regulation have also been defined in both cell biology and computational studies, each showing that Z-flipons are enriched in promoter regions [30,31,32,33,34,35]. The formation of GQ inside cells is also supported by genetic studies in which these structures accumulate in cells harboring loss-of-function helicase variants [36,37]. GQ binders have also been found in biochemical studies. The structure-specific interactions are validated using a non-GQ-forming oligonucleotide with the isoelectric substituent 7-deaza, 8-aza-guanosine bases as controls [38]. This base cannot form the Hoogsteen bases necessary to stabilize a GQ fold. Crystal studies reveal that the yeast B-DNA-specific RAP1 protein can also dock to GQ. Both interactions involve the same helix, but each through a different helical face. The B-DNA interaction occurs through the hydrogen-bonding face, whereas GQ docking depends upon the hydrophobic surface [39,40]. Roles for GQ have been validated for various biological outcomes [36,41,42,43,44].

Most recently, the binding modes of zinc-domain transcription factors (TFs) to DNA have been examined. These proteins constitute the largest and most ancient TF family with over 800 members. They are classically considered to bind only to B-DNA. However, predictive modeling studies with AF3 reveal that these proteins can also recognize ZNA and GQ [45]. Docking to Z-DNA was previously overlooked because the use of B-DNA competitors in binding assays likely masked interactions with Z-DNA. However, other experimental approaches have provided evidence for the interaction of zinc-finger domain (ZFd) transcription factors with both B-DNA and Z-DNA, with studies on zinc finger and BTB domain-containing protein 43 (ZBTB43) and Zinc finger and SCAN domain-containing protein 4 (ZSCAN4) revealing their role in suppressing Z-DNA formation by Z-prone alternating d(AC)_n_ microsatellites [46,47].

The finding helps resolve the long-standing speed-stability paradox posed by ZFd, which asks how proteins with multiple high-affinity B-DNA sites can rapidly scan the genome to find their cognate binding sites [48]. The targeting of Z-DNA reduces the search space to active regions of the genome, where sufficient energy is available to power the structural transition. The energy can be provided during the pioneering round of transcription by the displacement of nucleosomes (Figure 1E) or through the negative supercoiling produced by processive enzymes, such as RNA polymerase, which then powers the reinitiation of transcription (Figure 1F). The formation of Z-DNA flags nucleosome-free regions (NFRs). Once localized to an NFR, ZFd can then scan for a cognate binding site. The rapid off-rate from Z-DNA, resulting from the release of free energy upon the flip back to B-DNA, contrasts with the slow off-rate for ZFd bound to a specific sequence. The interaction between ZFd and the GQ formed by RNAs (rGQ) was also documented, potentially enabling the formation of complexes involved in RNA transcription, splicing, editing, and translation.

This study reveals that other B-DNA-specific transcription factor motifs are capable of docking to both ZNA and GQ. Furthermore, the proposed docking of homodimers onto Z-DNA can efficiently promote the local assembly of heterodimers with sequence-specificity for nearby promoters. The outcome depends on the relative protein levels of the potential partners, their sequence preference, and the regulatory complexes they seed. The models presented exploit the dynamic nature of flipons, where the exchange of energy for information potentiates the transition from one cell state to another.

## 2. Results

The use of flipons to regulate biological outcomes predates the evolution of sequence-specific binding factors [49]. Over time, more robust protein-centric controls eventually superseded the structure-based modulation of responses. These modern pathways were overlaid over old ones, adding features, rather than abandoning the innovations that were successful in the past. It was therefore of interest to investigate whether factors that act early in development dock to alternative flipon structures in a way that echoes past ontogeny. Consequently, we tested the interaction of Yamanaka factors [50] with rGQs and Z-DNA. These factors enable the reprogramming of differentiated cells to a pluripotential state and consist of the MYC proto-oncogene, Krüppel-like factor 4 (KLF4), OCT4 (encoded by *POU5F1*), and SOX2 (Figure 2).

The binding of Yamanaka factors to B-DNA is firmly established. MYC recognizes the Z-prone core sequence 5′-CACRTG-3′ (R = G/A) [51,52]. KLF4 interacts with the p300 histone acetyltransferase and regulates gene transcription by modulating histone acetylation, thereby inducing chromatin opening during cell reprogramming [53,54]. KLF4 recognizes the guanine-rich d(GGGTGGG) motif [55]. OCT4 binds through its POU domain to the d(ATTTGCAT) octamer motif [56]. SOX2 forms a complex with OCT4 on DNA to induce the readout of developmental genes, targeting nucleosome-bound sequences, such as d(TGTGGGAC) that differ from the canonical B-DNA binding sequences that are used later in development to regulate gene expression in differentiated tissue [57,58].

The binding of Yamanaka factors to flipon structures also has experimental support. For example, MYC binds GQ in vitro and RNA sequences capable of forming GQ in vivo [59,60]. SOX2 also experimentally binds a rGQ with nanomolar affinity [61]. AF3 was able to molecularly model the docking of both MYC to a telomeric rGQ substrate, consistent with the experimental results. The AF3 interaction involved the DNA-binding domain of MYC and the high-mobility group (HMG) domain. (panels A,C). Whether OCT4 or KLF5 dock to GQ has not yet been experimentally established. AF3 models reveal that OCT4 can dock to GQ through its POU homeobox domain (HD). When both OCT4 and SOX are present in the same AF3 model, the TFs engage rGQ through independent binding sites [55] (panel G). KLF4 also docks to the parallel-strand telomeric GQ through its second ZFd (panel I), as do the ZFd of other zinc-finger TFs [45].

All the Yamanaka proteins bound as homodimers in AF3 to Z-DNA. The docking of MYC to Z-DNA was promoted by the symmetrical dimethylarginine modification (dMeR) of R382. This class of modification is known to affect the oncogenicity of c-MYC, but currently available dMeR datasets do not cover the entire protein [62,63]. The PRmePRed tool (84.10% accuracy, 82.38% sensitivity, 83.77% specificity) predicts that the peptide (373-THNVLERQRRNELKRSFFA-391, score = 0.58) that contains the R382 residue is a site of arginine methylation [64]. The methyl group potentially forms a weak hydrophobic contact with the sugar carbons of guanosine, akin to the 5-methylcytosine modification that promotes Z-DNA formation [65,66]. The dMeR ammonium group strengthens the interaction by forming ionic bonds with the phosphate backbone. OCT4 also bound Z-DNA both as a monomer and as a dimer (panels G and H). Of note, reports indicate that the POU HD of OCT4 can interact with a methylated Z-prone dinucleotide motif, d(ATGCGCAT), which contains a CpG core [67]. Interestingly, SOX2 engaged Z-DNA as a monomer, but docking of two copies flipped the complex back to B-DNA (panels I and J). KLF4 bound to Z-DNA when one copy was present, in line with previous findings on the docking of zinc finger domains to Z-DNA (panel J) [45]. With two copies of KLF4, docking was to separate sites rather than through dimer formation.

TF with other sequence-specific B-DNA binding folds also docked to both GQ and Z-DNA (Figure 3). These include the master regulators of muscle development, Myoblast Determination Protein 1 (MYOD) and Myogenic Factor 5 (MYF5) [68]. Both bind through a bHLH motif to the canonical E-box CANNTG motif (N = any nucleotide) [69]. The MYOD1/MYF5 docks to anti-parallel GQ rather than the parallel telomeric GQ bound by the Hamanaka factors (Figure 3A). Docking of the MYOD1 homodimer to Z-DNA is facilitated by MYOD1 dMeR121 (116-MRERRRL-122) (PRmePRed scores for each arginine in the sequence RERRR = 0.670, 0,597,0.569, 0,623, respectively) (Figure 3B). Similarly, the MYF5 dMeR91 (88-MRERRRL-96) (PRmePRed score for first R = 0.660) binds as a homodimer to Z-DNA. However, the MYOD1/MYF5 heterodimers do not dock to Z-DNA under these conditions, even with dMeR modification (not shown). This disparity between homodimer and heterodimer binding to Z-DNA is found in AF3 models for other TFs, as described below. The TF protein C-ets-1 (ETS1) has a winged-helix-turn helix motif (wHTH) and recognizes a RCMGGAWGCY sequence motif (R = A/G, M = C/A, W = A/T, Y = C/T) [70,71]. The ETS domain is unique to animals and regulates a large number of cellular processes, with 28 family members in humans [72]. The protein binds to both the RNA telomeric GQ (Figure 3C) and Z-DNA without modification (Figure 3D). Cellular tumor antigen p53 (encoded by *TP53*) regulates an extensive set of target genes that control the cell cycle and apoptosis. The protein engages the p53 B-DNA response element through a highly structured domain, which has a half-site sequence of RRRCWWGYYY (R = A/G, S = G/C, W = A/T, Y = C/T), and frequently contains a CATG core [73]. This domain also binds to the telomeric GQ RNA sequence (Figure 3E), but, in the models tested, exhibited only limited affinity for Z-DNA (data not shown). The C-terminal unstructured domain also bound Z-DNA. Many docking geometries for this peptide were also observed, with the best one shown in Figure 3F. The C-terminal 26 residues have been previously reported to be essential for the expression of certain target genes that mediate TP53-induced growth arrest and apoptosis [74]. The FOS and JUN proto-oncogenes dimerize through a bZIP domain that recognizes a TGASTCA (S = G/C) motif in response to growth factor/receptor tyrosine kinase signaling [75]. The heterodimer engages the telomeric GQ (Figure 3G), with docking to Z-DNA facilitated by FOS dMeR115 (152-KCRNRRREL-161. PRmePRed score = 0.626) and JUN dMeR263 and dMeR270 modifications (256-ERKRMRNRIAASKCRKRKL-274 PRmePRed scores 0.633 and 0.626, respectively). Without modification, the FOS/JUN heterodimer bound only to B-DNA. Modified FOS (dMeR115) and JUN (dMeR263) homodimers also engage Z-DNA (not shown).

A different set of bZIP dimers involved in the antioxidant responses bound to Z-DNA as homodimers, but not as heterodimers, even with dMeR modification (Figure 4H). These proteins bind to antioxidant response elements (AREs) with a common core motif of 5′-TGACNNNGC-3′ [76]. The bZIP NRF2 (Nuclear factor erythroid 2-related factor 2, encoded by NFE2L2) dimerizes with small MAF (musculoaponeurotic fibrosarcoma) family members to induce genes that protect against oxidant stress. The response is suppressed by BACH1 (BTB domain and CNC homolog 1), which competes with NRF2 for the same proteins and the promoter ARE [77,78,79]. Homodimers with dMeR modifications all bound to Z-DNA, but preferred B-DNA when unmodified (Figure 4). The dMeR modifications for NRF2 and BACH1 are not at high-confidence sites and produce models that increase chain separation of the ZIP domains (NRF2 Figure 4A, 510-AQNCRKRKL-519, PRmePRed score < 0.5; Figure 4, BBACH, 571-AQRCRKRKLD-580, PRmePRed score < 0.5). In these cases, Z-DNA may promote the disassembly of these homodimers. Other factors, such as oxidative DNA base modifications, may also modulate the interaction. In contrast, the MAFF and MAFG dMeR sites are scored as high-confidence arginine methylation sites. (Figure 4C, MAFF, 66-ASCRVKRVC-74, PRmePRed score = 0.530; Figure 4D, MAFG, 66-ASCRVKRVT-74, PRmePRed scores = 0.557 and 0.508, respectively). Both these homodimers engage Z-DNA. As with MYOD1/MYF5, dMeR modified bZIP heterodimers preferred B-DNA over Z-DNA. Together, these findings are compatible with a model in which homodimers bind to Z-DNA, exchange chains to form heterodimers, and then dissociate due to a reduced affinity for Z-DNA (Figure 4H).

The MYC, MAX, and MAD1 proteins also bound to Z-DNA as homodimers when modified by dMeR, but not as heterodimers, even with more extensive dMeR substitutions (Figure 2B and Figure 4I–N). These proteins dock to the variable E-box CANNTG motif [69]. It was possible to swap the leucine-zipper and the Z-binding arms between these three proteins and still retain Z-DNA binding by dMeR14 modified homodimers. However, all possible heterodimer combinations formed with these domain swaps preferred B-DNA over Z-DNA. Replacement of individual residues in each arm revealed that multiple substitutions were required in both the leucine zipper and Z-DNA arms to achieve Z-DNA binding. Screening failed to identify other arginine, lysine, or serine modifications that favored an interaction with Z-DNA (results not shown).

In the models discussed so far, binding to Z-DNA relies on the ability of a dimer to promote the flip to Z-DNA. Not all Zα family members can induce the structural transition. For example, the vaccinia E3 protein binds to preformed Z-DNA but is unable to induce its formation. A structural rearrangement of the E3 tyrosine is required before Z-DNA engagement is possible [80]. To test whether heterodimers would preferentially bind preformed Z-DNA, AF3 models were run that included the well-characterized Zα domain (Figure 5A) [11]. The Zα domain stabilized Z-DNA in the models, enabling the determination of whether the heterodimers can engage the left-handed DNA conformer. In the presence of Zα, the NRF2 homodimer was able to dock to Z-DNA without dMeR modification (Figure 5B). Docking to Z-DNA of the NRF2-MAFF and BACH1-MAFF heterodimers was also observed (Figure 5C,D), indicating that these protein folds were able to bind preformed Z-DNA, but not promote the flip from B-DNA. Models of MAX-MAD1 and MAD1-MYC heterodimers made by swapping the leucine zipper and Z-DNA arms were also able to engage Z-DNA in the presence of Zα (Figure 5E–H).

## 3. Discussion

The approach taken here to investigate such interactions represents a novel strategy to overcome the limitations and time constraints associated with current experimental techniques. The reproducibility of interactions across many models and proteins increases confidence in the robustness of the findings, as does the recent experimental validation of models for the docking of general transcription factor E (TFEα encoded by *GTFE1*) to Z-DNA and the ADAR Zβ domain to GQ, as described in the methods. The analyses demonstrate that the bHLH and bZIP folds can accommodate the Z-DNA helix, in addition to binding B-DNA. Such interactions have never been previously assessed at the molecular level. The in silico domain swaps show the interchangeability of the dimerization and Z-DNA-binding domains of bHLH proteins (Figure 3 and Figure 4). The models enable the exploration of protein modifications that can dynamically alter the B-Z DNA equilibrium in a cell. Previously, only DNA modifications, such as methylation, oxidation, and base adducts, were known to modulate Z-DNA formation [65,81,82,83].

The methylarginine modifications described here warrant further experimental exploration, given their effectiveness in models of switching the binding preferences of homodimers from B- to Z-DNA, and their overlap with predicted sites of arginine methylation. There are specific inhibitors of arginine methyltransferase under development that could be used to examine how this modification affects Z-DNA formation in vivo. Techniques also exist for the targeted recoding of the arginine residues involved [84,85,86]. Genetic approaches may also help determine the role of dMeR in Z-DNA formation. For example, the human MYF5 variant R95C, which affects the dMeR modeled here, diminishes MYF5 binding and nuclear localization and produces a clinical disorder characterized by congenital ophthalmoplegia, scoliosis, and vertebral and rib anomalies [87]. The modeling results motivate the derivation of additional experimental data to examine the role of Z-DNA formation in such diseases.

The AF3 models also revealed unexpected differences between the binding of homodimers and heterodimers to Z-DNA. Homodimers modified by dMeR bind to Z-DNA. In contrast, heterodimers first require Zα to stabilize this structure, suggesting that the interactions are of lower affinity. However, the dimerization interface of both homo- and heterodimers is weak and permissive of chain exchange. Heterodimers with varying B-DNA sequence specificity can then rapidly form by the dissociation and reassociation of existing dimers. The heterodimer mix then varies in proportion to the cellular concentrations of each homodimer, with subunit exchange facilitated by the localization of homodimers to the Z-DNA surface, as shown here. Notably, the reduced affinity of heterodimers to Z-DNA diminishes the reverse reaction in which the exchange of chains instead reforms homodimers.

The binding of a heterodimer to a cognate B-DNA sequence stops any further exchange by preventing subunits from dissociating, thereby canalizing downstream events [88]. Notably, bZIP heterodimers recognize a large number of different sequences, as reflected in the degenerate motif 5′-TGACNNNGC-3′ bound by this family of TFs [76]. A subset of these heterodimers even recognizes more than one sequence combination [89]. Currently, it is unclear from existing B-DNA-based models how each heterodimer identifies the location of its preferred target. The concentration of homodimers on the ZNA surface (either Z-DNA or Z-RNA) helps pair different bZIP chains by promoting the disassembly of homodimers and the assembly of heterodimers (Figure 4E–H). The output of this assembly line depends on the homodimers produced by a cell. Some dimer combinations will not recognize a local sequence motif and will repeat the disassembly and reassortment cycle. Others will be stabilized by binding to DNA, but not impact promoter function [90,91,92]. Still others will bind B-DNA and either enhance or suppress gene expression depending on the complexes they seed. Those heterodimers that promote transcription will generate negative supercoiling, which can then power ANS formation by nearby flipons. Those sequences that flip to Z-DNA will provide an additional surface for assembling additional heterodimer variants. Those heterodimers with a higher affinity for a local B-DNA sequence will eventually be enriched in the region. This process can regulate tissue development by discovering the heterodimer combinations most adaptive to environmental inputs and cell state, without requiring their precise genetic specification.

The high-resolution models presented here are amenable to experimental evaluation. The findings will extend our understanding of what is possible and improve our knowledge of why previous laboratory work failed to uncover such mechanisms. One prevalent bias is that methods are optimized to show protein binding to B-DNA, and not to other possible flipon conformations. For example, the inclusion of competitor B-DNA into assays may interfere non-specifically with the binding of a domain to GQ and Z-DNA, just as it does with its sequence-specific recognition of B-DNA. Many structural studies optimize for the lowest-energy conformation. In most cases, the crystallization conditions used are based on the assumption that a protein binds only to B-DNA. Designing studies to detect the interaction between TFs and ANS will yield new insights. The design of in vivo, structure-sensitive reporter probes to track ANS in real time will also significantly advance our knowledge of flipon biology.

The binding of TF to both B-DNA and GQ is not unexpected, as some previous studies provide precedent. There is structural evidence that the yeast RAP1 protein docks to both B-DNA and GQ via different faces of the same α-helix [39,40]. Biochemical evidence suggests the same possibility for other proteins [59,93,94,95,96]. The existence of TFs that engage both B-DNA and Z-DNA is also not unexpected, despite previously expressed opinions to the contrary [10,97]. Rather than being a rare event, the current analysis provides evidence that TFs commonly engage multiple nucleic acid conformations. Of the proteins tested, only TP53 engages GQ and Z-DNA through different domains, suggesting that either TP53 can simultaneously engage different flipon conformations, or that TP53 binding to each structure can be independently regulated. An earlier study also indicates that TP53 binds triplexes, further supporting a role for flipons in the cell fate decisions enforced by TP53 [98].

The dynamic nature of transcriptional complex assembly is supported by recent evidence from single-cell immunofluorescent studies. RNA polymerase II transcriptional clusters have an average lifetime of 5.1 (±0.4) seconds and disassemble rapidly [99,100]. The same is true for the engagement of SOX2 and OCT4 near transcription start sites (TSS). Both have residence times of approximately 10 s, with only 20–50% of each bound to chromatin at any given time. In embryonic stem cells, 64 ± 7.8% of SOX2 is engaged with euchromatin and 16 ± 4.5% is present in heterochromatin. The outcomes reflect the docking of SOX2 and OCT4 to a site near a nucleosomal entry/exit site rather than to a site within the nucleosomal dyad [101]. 

Pioneering TFs such as SOX2 and OCT4 play essential roles in establishing DNA topology during development. They induce partial or complete ejection of nucleosomes, thereby unwrapping the DNA to produce negative supercoiling [102,103]. The energy captured in these underwound duplexes is sufficient to drive the flip from B-DNA to an ANS [104,105]. In the case of OCT4, the POU HD binds a non-canonical sequence that is accessible on the nucleosome surface and different from the canonical sequence recognized in nucleosome-free B-DNA. During docking, the POU HD forms a wedge that unwraps ~25 bp of DNA at the nucleosome entry site, eventually leading to nucleosome ejection and Z-DNA formation [58]. The recognition of Z-DNA localizes cell-specific TFs to the region. The TFs can then scan for a B-DNA-cognate binding site. Honing in on Z-DNA helps reduce the search space, speeding TF engagement. The off-rate from Z-DNA is likely to be as fast as the energy released when Z-DNA reverts to B-DNA will drive dissociation, speeding TF scanning of the genome. In contrast, the off-rate for sequence-specific binding to B-DNA is much slower, favoring more stable interactions that risk trapping TFs at non-functional sites throughout the genome.

Once established, reinitiation of transcription follows different kinetics from the pioneering round. The process requires recruiting RNA polymerase, pausing the polymerase to ensure the transcriptional machinery is fully loaded, and then releasing it to begin transcript elongation. Each cycle happens rapidly. Typically, promoters are ‘active’ for 3.0 ± 1.0 min, with a measured ‘burst’ size (RNA/active period) of 1.5 ± 0.5 transcripts [106]. The burst frequency, rather than the burst size, varies with the residence time of the TFs [107]. The length of inactive periods between bursts is highly variable. Somewhere between 1–10% of paused complexes enter productive elongation [108,109]. The cycle involves several distinct steps, during which flipons can actuate different outcomes (Figure 1F) [105]. Following the initiation of RPOL transcript elongation, the formation of Z-DNA is powered by the negative supercoiling generated. The release of negative supercoils as Z-DNA reverts to B-DNA offsets the positive supercoiling that stabilizes the preinitiation complex [110], leading to its disassembly [105]. Z-DNA also promotes the reengagement of Transcription Factor E, a necessary step for the reinitiation of the flipon cycle [111]. Z-DNA formation also facilitates scanning of promoter DNA by local TFs for a cognate binding site. Following TF docking and assembly of a promoter complex, a short RNA transcript is produced, and the RPOL temporarily enters a paused state [112]. The promoter is then locked and loaded until receiving the signal to fire.

The formation of GQ appears to be part of the transcription cycle and may depend on a brief round of antisense transcription. This process would dislodge TF bound to upstream sequences, enabling GQ formation by either the antisense RNA transcript produced or by the non-template DNA strand displaced during transcription. With repeated rounds of antisense transcription, the antisense RNA GQ becomes more abundant. However, that RNA subset is usually rapidly cleared by the RNA exosome. Non-template DNA can potentially embed many flipons capable of forming GQ. Docking of TF to GQ, either RNA or DNA, rather than to B-DNA motifs, then explains the High Occupancy Sites (HOT) identified by the ENCODE Consortium. These HOT sites engage multiple TFs, even when their binding motifs are absent [113]. TF-associated proteins will also localize to these sites, delivering the helicases needed to resolve GQ once the transcriptional burst is triggered. The HOT complexes then facilitate promoter reset and the reengagement of B-DNA-specific transcription factors. The accumulated non-productive transcripts from genes, enhancers, and promoters can also adopt alternative flipon conformations. They may enable the local retention of TFs and other factors for use in the next transcription cycle. They may also prevent the reassociation of a particular TF with DNA, thereby allowing it to be replaced by another TF or by a different complex [114].

The results presented here reveal that transcription factor motifs can potentially engage multiple alternative flipon conformations. The models motivate further experimental validation. The models provide functional endpoints for these studies, allowing a clear readout of their validity. That said, the current experimental evidence supports an essential role for flipons in regulating the dynamic readout of genetic information from DNA. The repeat motifs that underlie these alternative structures are simple. They set the stage for nature to strut its highly evolved codonware.

## 4. Methods

AFV3 is not explicitly trained to recognize Z-DNA or other flipon structures, nor their interactions with protein motifs. However, large neural networks often contain subnetworks that, when properly initialized, can perform well on specific tasks [115]. Empirically, it was found that AF3 can be nudged by setting input conditions and specifying the model seed, enabling exploration of the interactions between Z-flipon folds and proteins [49]. Optimal starting conditions were calibrated using known ZNA binders (Zα and TFEα)) and previously described GQ binders [59,111]. It was found that the AF3 default scoring of these models did not always assign high scores to the well-validated interaction between Zα and Z-DNA. Instead, a better indication of their significance was provided by manual inspection of the models produced. Models with many well-oriented binding interactions and no steric clashes were then selected for further evaluation. This approach was judged reasonable as the AF3 algorithm does not explicitly score bonding interactions. To test the novel predictions made with AF3, we have used molecular dynamics simulations (MDS) for preliminary validation. We find that the selected AF3 models are robust. In contrast, other modeling approaches that employ alternative docking algorithms are prone to failure in MDS, reflecting the sensitivity of this deterministic method to errors in the initial model. Using this strategy, we have validated the AF3-modeled interaction between the Zβ domain of ADAR and GQ using MDS. The result has subsequently been confirmed by NMR [116,117]. The MDS also overcomes another limitation of AF3, which captures only a snapshot of the interaction. MDS allows modeling of the dynamic transitions involved. Established physical-chemical principles allow inference of a protein’s relative affinity for B-DNA and Z-DNA based on the bonding scheme present in AF3 models. For example, the slow off-rate of Zα from Z-DNA is due to the face-on edge contact of tyrosine with the C8 of a purine base, not solely on ionic or hydrogen bonds [13,118]. In general, the release of the potential energy from the reversion of Z-DNA to B-DNA will favor a fast off-rate [119].

The following parameters were used here to model the interaction of transcription factors with Z-DNA: a d(CG)8 repeat in the presence of 12 Mg^2+^, 6 K^+^, and 4 Zn^2+,^ and seed = 1076865862. A screenshot of the conditions used to initiate the model is given in the Appendix A. For modelling G-quadruplex, the parallel strand r(GGGUUAGGGUUAGGGUUAGGG) telomere sequence and the anti-parallel C9ORF72 r(GGGGCCGGGGCCGGGGCCGGGG) in the presence of 3xK^+^, and seed = 528133945. Although proteins docked to both DNA GQ (dGQ) and RNA GQ (rGQ), higher-order stacked GQ structures were formed best with RNA folds. As before, model selection for rGQ and Z-DNA was based on bonding scheme and steric fit. The metal-ion interactions that nudge AF3 to fold the desired nucleic acid structure were ignored in these assessments. This approach is conservative, as metal interactions may help stabilize the complexes under physiological conditions and, consequently, contribute significantly to their stability. AlphaFold V3 was also used to model the effects of various amino acid adducts and substitutions on TF binding. These predictive models generate several outcomes. First, they increase knowledge of what interactions are possible when modeled at atomic resolution. Second, they allow an assessment of how various mutations and domain swaps affect binding. Third, they allow modeling of different amino modifications.

## 5. Conclusions

This paper questions whether transcription factors (TF) interact exclusively with B-DNA rather than other DNA structures. The analysis presented here is based on predictive modeling and provides evidence that TFs also bind to alternative DNA conformations, such as left-handed Z-DNA and G-quadruplexes. These alternative structures are encoded by repeat sequences called flipons that perform various roles in the cell. Alternative flipon structures flag active genes that generate sufficient energy to power the transition. They enable TF to quickly scan the genome to identify cognate binding sites, thereby facilitating rapid responses. By cycling conformation, flipons also enable promoter reset and reinitiation of transcription. In other cases, flipons may promote the local assembly of heterodimeric transcription factors from homodimers. This process finds the monomer combination that binds a control locus with sufficient affinity to regulate its activity. This reassortment of monomers is facilitated by weak dimerization interfaces that allow homo- and heterodimers to dissociate and then reassociate to form heterodimers with a different B-DNA sequence specificity. The binding of homodimers to alternative DNA structures increases their local concentration and facilitates exchanges that underlie the formation of new heterodimer combinations. This process is likely regulated in several ways, most notably by protein modifications, such as arginine methylation, and DNA modifications, such as methylation and oxidation, that alter the relative affinities of homo- and heterodimers for each DNA structure.

## Figures and Tables

**Figure 1 ijms-27-00272-f001:**
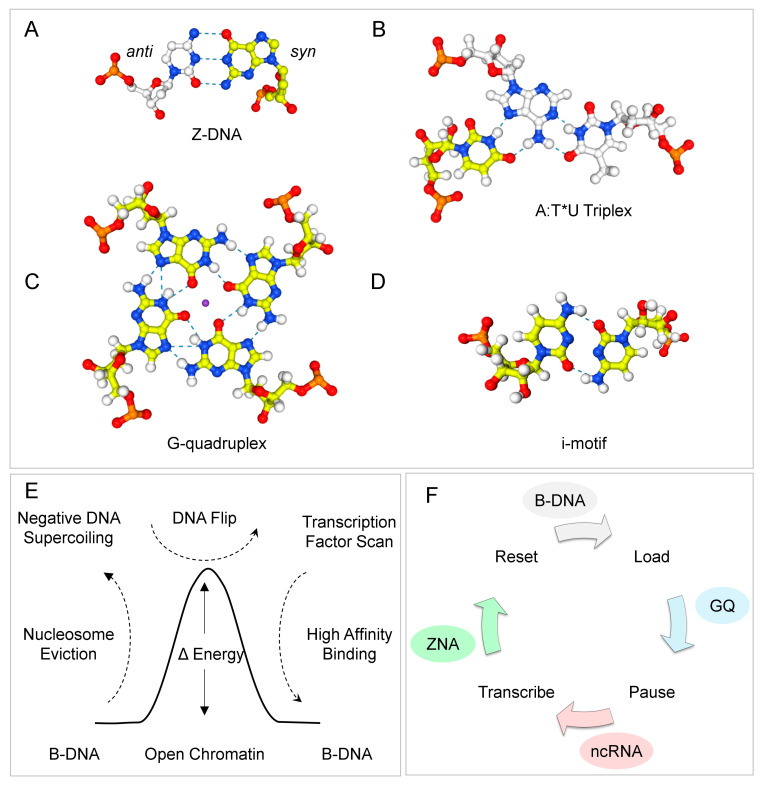
Flipons adopt alternative conformations by rotation of nucleobases around the glycosidic bond from the anti (shown in white) to the syn conformation (shown in yellow). (**A**) The ZNA motif consists of an alternating purine-pyrimidine repeat: the purine bases in syn and the pyrimidine bases in anti. The pattern produces a left-handed duplex with a zig-zag backbone. The dotted lines represent hydrogen bonds. (**B**) A triplex formed by Hoogsteen basepairing (indicated by “*”) between the third strand and the right-handed Watson–Crick duplex (basepairing is represented by “:”). (**C**) G-quadruplexes (GQ) are built by stacking a guanosine tetrad (G4) that is formed by Hoogsteen base pairing of four guanine bases. The central GQ core can accommodate a metal ion, such as potassium or sodium. (**D**) An i-motif intercalates pairs of cytosine base-pairs to form a four-stranded structure. (**E**) During the pioneering round of transcription, the eviction of nucleosome releases sufficient negative supercoiling to power the transition of Z-flipons from the B-DNA to the Z-DNA conformation. Recognition of Z-DNA by TF enables the rapid discovery of its cognate binding sites. The energy released as the flipon reverts to B-DNA then powers the assembly of complexes at the locus. (**F**) Formation of Z-DNA is also powered by the negative supercoiling generated by an elongating polymerase. The energy captured by Z-DNA can drive the reset and reinitiation of promoters. At the same time, the formation of GQ likely plays a role in the pausing and release of an assembled polymerase complex before the next burst of transcription.

**Figure 2 ijms-27-00272-f002:**
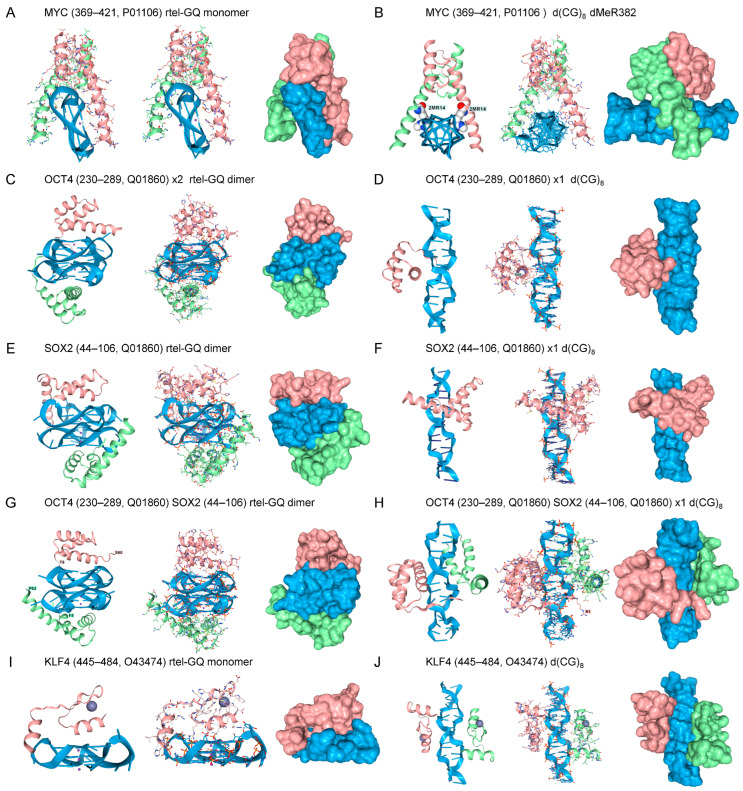
Docking of Yamanaka factors and the master regulator to GQ (**A**,**C**,**E**,**G**,**I**) and Z-DNA (**B**,**D**,**F**,**H**,**J**). The amino acids and their corresponding UniProt Accession numbers are listed in the title of each panel. The position of N3, N4-Dimethyl-L-arginine (dMeR) substitution in MYC is labeled. In each panel, and in subsequent figures, three renditions are shown: a cartoon representation, a combined cartoon and licorice representation with bonding, and a surface representation for each component. Z-DNA and GQ are colored blue, each protein domain is colored green or fuchsia, and the dMeR is shown in a ball-and-stick representation with white carbons.

**Figure 3 ijms-27-00272-f003:**
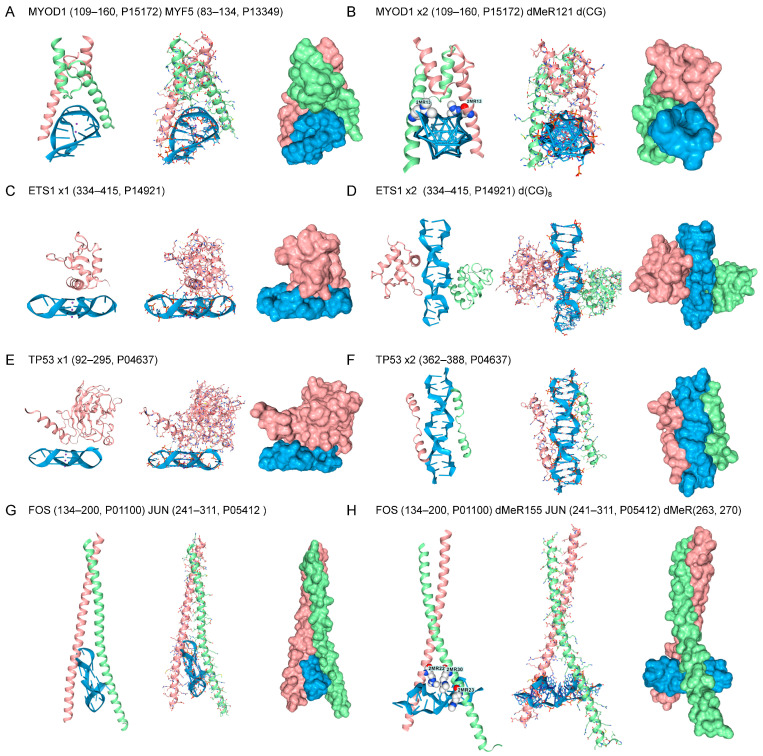
Other TF Motifs. (**A**) The MYOD1-MYF5 bHLH heterodimer bound to GQ. (**B**) The dMeR modified MYOD1 homodimer bound to Z-DNA. (**C**,**D**) ETS1 docks to Z-DNA and to a parallel-stranded telomere rGQ (**E**,**F**) DNA repair protein TP53 bound to rGQ. (**G**,**H**) Leucine zipper FOS-JUN heterodimer bound to Z-DNA and rGQ.

**Figure 4 ijms-27-00272-f004:**
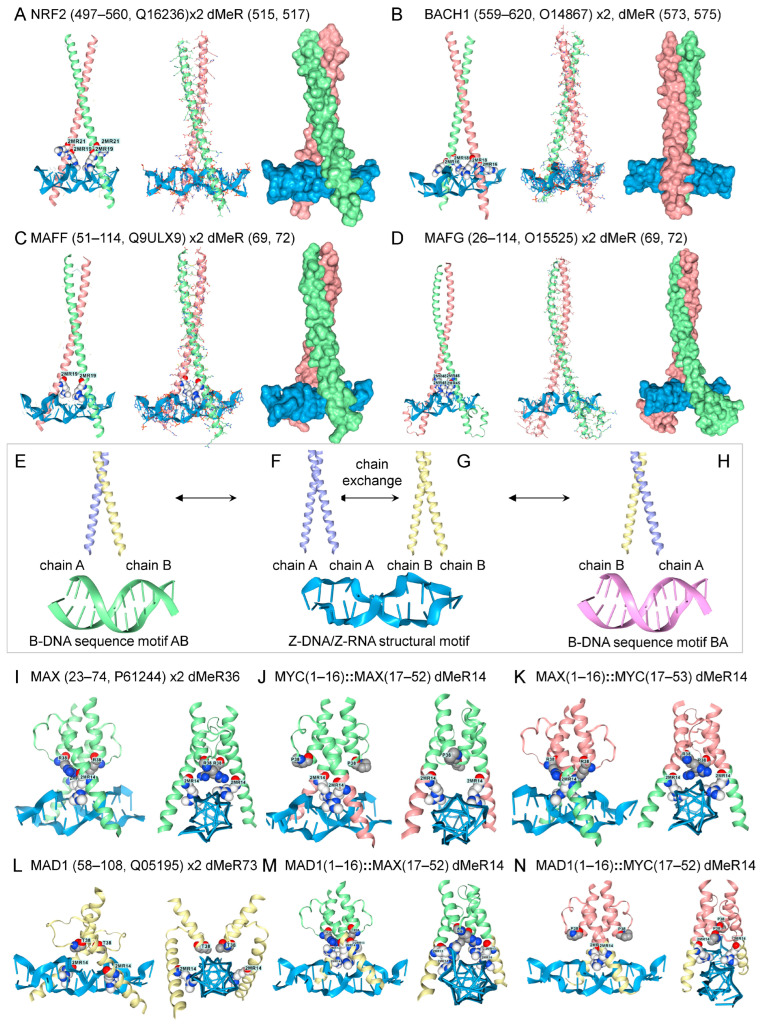
Mix and match bZIP and bHLH dimers. (**A**) NRF2 and (**B**) BACH1, and their binding partners (**C**) MAFF, (**D**) MAFG, dock to Z-DNA as homodimers. (**E**) Docking of homodimer pairs onto Z-DNA may facilitate the formation of heterodimers. The heterodimer AB, generated by exchange of chains between homodimer AA (**F**) and homodimer BB (**G**) on ZNAs, could bind a different B-DNA motif than the reciprocal pairing BA (**H**). The MAX and MYC homodimers bind to Z-DNA, but the heterodimer does not. (**I**–**N**) The bHLH motif and the ZNA-binding arms can be exchanged between MAX and MYC, while retaining ZNA binding. The colors indicate the segments that were swapped, and the numbers give the residues exchanged. The numbers are for the construct, not for the position in the protein. The ‘::’ indicates a fusion. The MAX and MAD1 homodimers also bind to Z-DNA (**I**,**L**). The bHLH arms of MAD1, MAX, and MYC can also be combined with the MAD1 ZNA-binding arms without affecting binding to ZNA. However, all six possible heterodimer combinations preferentially dock to Z-DNA rather than to B-DNA.

**Figure 5 ijms-27-00272-f005:**
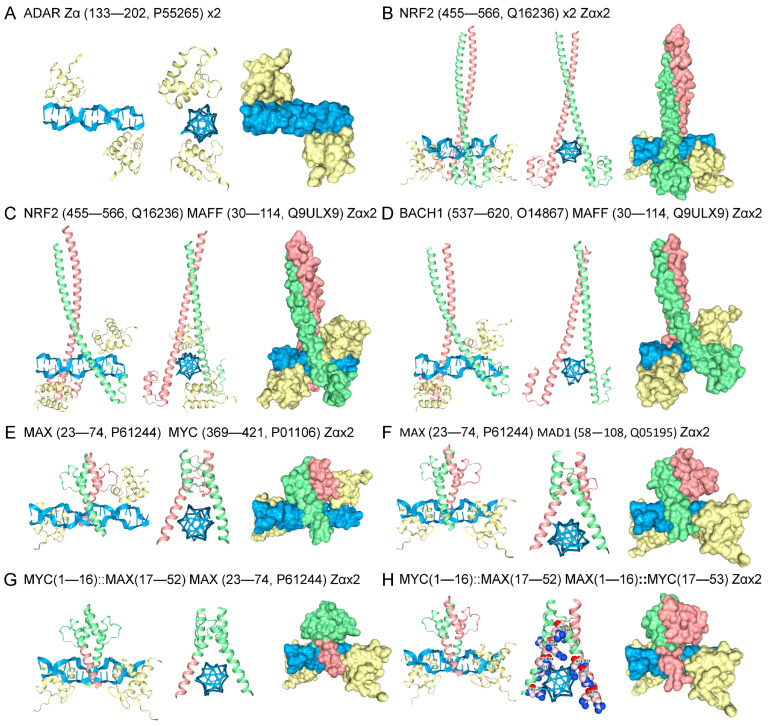
Binding of Heterodimers to Preformed Z-DNA. (**A**) The Zα domain flips d(CG)8 to the Z-DNA conformation. (**B**) Including Zα in models removed the requirement of dMeR modification to promote Z-DNA binding by NRF2. This approach provided a way to directly assess whether bZIP and bHLH heterodimers can dock to Z-DNA without the need to induce its formation. Binding of (**C**) NRF2- and (**D**) BACH1-MAFF heterodimers to Z-DNA. Engagement of Z-DNA by the (**E**) MYC-MAX and (**F**) MAX-MAD1 heterodimers. The Zα is colored yellow, Z-DNA is blue, and each monomer is either green or fuchsia. The binding of the mix-and-match heterodimers to Z-DNA was also tested. (**G**) The MAX heterodimer, where a Z-DNA binding arm was replaced with one from MYC. The colors indicate the segments that were swapped, and the numbers give the residues exchanged. The ‘::’ indicates a fusion. (**H**) A pairing of MAX and MYC bHLH motifs, with the Z-DNA binding arms from MAX and MYC swapped between the monomers.

## Data Availability

PDB files for all the Figure Panels are supplied as Appendix A. The conditions used for AF3 to determine the structures are given by the file titles.

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
