# Peer review of "Control of Gene Expression by Proteins That Bind Many Alternative Nucleic Acid Structures Through the Same Domain"

_ijms, 2025, doi:10.3390/ijms27010272_

Round 1
Reviewer 1 Report
Comments and Suggestions for Authors
I appreciate the opportunity to review this article you sent to me. I am writing to provide my feedback on the article titled “Control of Gene Expression by Proteins that Bind Many Alternative Nucleic Acid Structures Through the Same Domain” submitted to the International Journal of Molecular Sciences (IJMS). As a reviewer, I have carefully reviewed the article and would like to share my insights for your consideration. My major comments and suggestions are as follows:
Major Concerns:
- The author has overlooked the major limitation that AlphaFold V3, is a static structure-prediction tool. It cannot account for the inherently dynamic and reversible behavior of flipons. In physiological conditions, different interactions are energy-dependent and transition between B-form DNA and alternative conformations such as Z-DNA or G-quadruplexes. Therefore, a single static model cannot capture the kinetics, thermodynamics, or regulatory features that govern these conformational equilibria and thus provides only a partial view of their functional properties.
- The author has used only a couple of heuristic “nudging” choices, including narrowly defined ion concentrations and a few random seeds, which raises questions about the robustness and reproducibility of the results. Such parameter sensitivity may limit the broader applicability of the conclusions.
- The author has not provided any quantitative estimates of binding affinity or interaction energetics. This limits the ability to assess the strength, specificity, or biological relevance of the predicted complexes.
- These predictions lack experimental validation and are difficult to interpret with confidence, as they rely solely on static conformations generated without quantitative binding affinity estimates, without modern deep-learning–based evaluation, and with only a narrow set of seed sequences and ion conditions.
Thank you for considering my feedback. Please feel free to contact me if you require any further clarification or information.
Author Response
Thanks for your review and the opportunity to address your comments.
Comment 1: The author has overlooked the major limitation that AlphaFold V3, is a static structure-prediction tool. It cannot account for the inherently dynamic and reversible behavior of flipons.
Response 1: The methods section has been expanded to address these questions. We only claim that AF3 provides snapshots of the interaction. These snapshots correspond to minima in the energy landscape. Also added was the use of MDS in subsequent work to examine the transition from one state to another. The example is given of how we used AF3, then MDS then NMR to validate the approach in the case of the ADAR Zβ binding to G-quadruplexes
Comment 2 The author has used only a couple of heuristic “nudging” choices, including narrowly defined ion concentrations and a few random seeds, which raises questions about the robustness and reproducibility of the results.
Response 2. The results are robust, as shown by the diversity of models shown and by examination of the PDB files describing the structures obtained. The “reason for “heuristic” nudging is also explained in the methods section. It relates to how the modeling works. IN AF3 and other reinforcement models, a number of subnets are trained that are optimal for different tasks. It is possible to access a particular subnet by using different starting conditions and seeds. The hypotheses generated can then be tested using MDS and experimental studies as we did fir Zβ.
Comment 3, The author has not provided any quantitative estimates of binding affinity or interaction energetics. This limits the ability to assess the strength, specificity, or biological relevance of the predicted complexes.
Response 3. These are experimental questions. The techniques are well known and have been implemented by many labs to investigate the binding of Zα to Z-DNA, Z-RNA and B-DNA.
Comment 4. These predictions lack experimental validation and are difficult to interpret with confidence, as they rely solely on static conformations generated without quantitative binding affinity estimates, without modern deep-learning–based evaluation, and with only a narrow set of seed sequences and ion conditions.
Response 4. I think the predictions are easy to interpret, as they open up a whole new field of biology. They question current models dominated by the sequence-specific binding of proteins to B-DNA. Many key questions are not ignored. We recently proposed a resolution to the paradox of how zinc finger domain proteins enable fast response when their affinity for B-DNA is so high by using a flipon based approach. This paradox has existed for over 50 years, but without the proper conceptual framework, it has proven intractable. The hope is that this paper will also help address the narrow set of concepts that has limited the experimental interrogation of the role of dynamic DNA conformations in cellular biology.
Reviewer 2 Report
Comments and Suggestions for Authors
Please find attached

Author Response
Thanks for taking the time to review the paper and for your helpful suggestions.
Reviewer 1
Comment 1. While the hypothesis is provocative and potentially significant, the manuscript relies entirely on in silico predictions without wet-lab validation, a limitation that must be addressed more rigorously in the Discussion.
Response. The comment is valid, but does not invalidate the in silico findings. The approach has already proven useful in many studies, providing insights that cannot be gleaned from the current experimental data. Moreover, with reasons to perform experiments to test the binding of transcription factors to alternative conformations, such data will never be produced.
In response to the comment, I have highlighted existing studies that support the in silico work by expanding the sections in the earlier manuscript to describe these findings more fully. I have also added the NMR experimental validation that is under review for predictions using this approach for the docking of the ADAR Zβ to G-quadruplexes. We also have confirmation for the binding of TFEα to Z-DNA and some very nice MDS work showing how the interaction opens up the helix. However, this manuscript is not yet submitted.
Comment 2: The Introduction provides a solid background on flipons and Z-DNA biology but lacks critical context regarding computational tools and broader TF biology. The introduction fails to discuss the current limitations or confidence levels of AF3 specifically for non-canonical DNA structures.
Response 2: I have expanded the methods section to explain how the model is used, and the empirical nature of the parameters employed for modeling to overcome limitations in AF3 and its scoring metrics. The results cover a wide range of transcription factors and sequence variations, yet remain consistent with one another. I also show that the sites of dMeR modification overlap those predicted in the PRmePRed algorithm, even though this tool was not used to guide the study. Furthermore, the models are presented as PDB files to enable visualization of the predictions at atomic resolution. This information allows biologists to make their own assessment of the findings, something that many find difficult to do with other computational approaches.
Comment 3: The transition to discussing Yamanaka factors (MYC, KLF4, SOX2, OCT4) is abrupt.
Response 3. I have rewritten this section to introduce the rationale better and describe the biology.
Comment 4: The text mentions ADAR "squelching" ZBP1 activation, but this specific mechanism is dense and may alienate readers not familiar with the author's previous work on Z-RNA. A clearer explanation of how this relates to general TF binding would improve flow.
Response 4. I have also expanded this section.
Comment 5. The Discussion is speculative and lacks necessary caveats regarding the methodology. The most critical omission is the lack of wet-lab validation.
Response 5: This paper is not a wet lab paper. It demonstrates the use of AF3 to perform computational experiments to test various hypotheses and triage those that are uninformative. For example, the domain swap and mutational experiments performed are both labor- and cost-intensive as wet-lab experiments. The computational analysis provides insights into how the DNA-binding and dimerization domains evolve independently to specify interaction with Z-DNA. For those interested and with the reagents available to investigate this question, the domain swaps most likely to be informative have now been defined. I am not sure that these experiments would be performed without having the rationale provided by the computational analyses. A similar comment for the dMeR modifications.
Comment 6; The discussion should explicitly frame this work as "theoretical" or "predictive."
Response 6. Thanks for the suggestion. I now refer to the approach as predictive modeling. I believe this method provides valuable insights that go far beyond the wet-lab-only approaches. As more data accumulates and AF4, AF5, or AlphaFlipon models are elaborated, the predictive modelling approach will only become more informative.
Comment 7. The Discussion lacks evidence that these specific residues (e.g., MYC R372, FOS dMeR115) are actually methylated in vivo at the time of transcription. Without proteomic evidence, this mechanism remains purely hypothetical. There is no discussion of the AlphaFold confidence scores (pLDDT or PAE plots) for these specific models. In structural biology predictions, discussing the model confidence—specifically for the DNA-protein interface—is mandatory to assess reliability
Agreed. I commented above that I the sites of dMeR modification overlap those predicted in the PRmePRed algorithm, a SVD model that is based on experimental data. I also expanded the methods section, as described above, to address model scoring. If I or anyone else had a lab well-funded enough to perform the experimental validation, I think the paper would be submitted to Nature or Science. Such papers will arise in the future once there is enough consensus to fund and perform these experiments. This paper is to help provide the rationale that is currently lacking. The reviewer clearly has some insight into the value of performing these experiments and the best approaches to do that. Hopefully, the reviewer will pick up the gauntlet they have thrown and test the findings experimentally.
Round 2
Reviewer 1 Report
Comments and Suggestions for Authors
I find the authors’ replies satisfactory, and the revisions adequately address the major concerns raised in my initial review. In particular, the limitations of the computational approach are now more clearly acknowledged, and the interpretation of the predicted interactions has been appropriately refined. Overall, these changes improve the clarity and strength of the manuscript.
Reviewer 2 Report
Comments and Suggestions for Authors
The author has incorporated the suggestions to the best of his capacity.